# Capsules for Object Segmentation

**Rodney LaLonde**
Center for Research in Computer Vision
University of Central Florida
lalonde@knights.ucf.edu

**Ulas Bagci**
Center for Research in Computer Vision
University of Central Florida
bagci@crcv.ucf.edu

## Abstract

Convolutional neural networks (CNNs) have shown remarkable results over the last several years for a wide range of computer vision tasks. A new architecture recently introduced by Sabour et al. [2017], referred to as a capsule networks with dynamic routing, has shown great initial results for digit recognition and small image classification. The success of capsule networks lies in their ability to preserve more information about the input by replacing max-pooling layers with convolutional strides and dynamic routing, allowing for preservation of part-whole relationships in the data. This preservation of the input is demonstrated by reconstructing the input from the output capsule vectors. Our work expands the use of capsule networks to the task of object segmentation for the first time in the literature. We extend the idea of convolutional capsules with *locally-connected routing* and propose the concept of *deconvolutional capsules*. Further, we extend the masked reconstruction to reconstruct the positive input class. The proposed convolutional-deconvolutional capsule network, called **SegCaps**, shows strong results for the task of object segmentation with substantial decrease in parameter space. As an example application, we applied the proposed **SegCaps** to segment pathological lungs from low dose CT scans and compared its accuracy and efficiency with other U-Net-based architectures. **SegCaps** is able to handle large image sizes ($512 \times 512$) as opposed to baseline capsules (typically less than $32 \times 32$). The proposed **SegCaps** reduced the number of parameters of U-Net architecture by 95.4% while still providing a better segmentation accuracy.

## 1   Introduction

Object segmentation in the medical imaging and computer vision communities has remained an interesting and challenging problem over the past several decades. Early attempts in automated object segmentation were analogous to the if-then-else expert systems of that period, where the compound and sequential application of low-level pixel processing and mathematical models were used to build-up complex rule-based systems of analysis. Over time, the community came to favor supervised techniques, where algorithms were developed using training data to teach systems the optimal decision boundaries in a constructed high-dimensional feature space. In computer vision fields, superpixels and various sets of feature extractors such as scale-invariant feature transform (SIFT) (Lowe [1999]) or histogram of oriented gradients (HOG) (Dalal and Triggs [2005]) were used to construct these spaces. Specifically in medical imaging, methods such as level sets (Vese and Chan [2002]), fuzzy connectedness (Udupa and Samarasekera [1996]), graph-based (Felzenszwalb and Huttenlocher [2004]), random walk (Grady [2006]), and atlas-based algorithms (Pham et al. [2000]) have been utilized in different application settings.

In the last few years, deep learning methods, in particular convolutional neural networks (CNNs), have become the state-of-the-art for various image analysis tasks. Specifically related to the object segmentation problem, U-Net (Ronneberger et al. [2015]), Fully Convolutional Networks (FCN)

1st Conference on Medical Imaging with Deep Learning (MIDL 2018), Amsterdam, The Netherlands.

(Long et al. [2015]), and other encoder-decoder style CNNs (*e.g.* Mortazi et al. [2017a]) have become the desired models for various medical image segmentation tasks. Most recent attempts in the computer vision and medical imaging literature utilize the extension of these methods to address the segmentation problem. Since the success of deep learning depends on finding an architecture to fit the task, currently several researchers work on designing new and more complex deep networks to improve the expected outcome. This naturally brings high number of hyperparameters to be configured, makes the overall network too complex to be optimized.

**Drawbacks of CNNs and how capsules solves them**

The CNNs, despite showing remarkable flexibility and performance in a wide range of computer vision tasks, do come with their own set of flaws. Due to the scalar and additive nature of neurons in CNNs, neurons at any given layer of a network are ambivalent to the spatial relationships of neurons within their kernel of the previous layer, and thus within their effective receptive field of the given input. Recently Sabour et al. [2017] introduced the idea of *capsule networks*, where information at the neuron level is stored as vectors, rather than scalars. These vectors contain information about:

1. spatial orientation,
2. magnitude/prevalence, and
3. other attributes of the extracted feature

represented by each capsule type of that layer. These sets of neurons, henceforth referred to as capsule types, are then "routed" to capsules in the next layer via a *dynamic routing algorithm* which takes into account the agreement between these capsule vectors, thus forming meaningful part-to-whole relationships not found in standard CNNs.

**The overall goal** of this study is to extend these capsule networks and the dynamic routing algorithm to accomplish the task of object segmentation. We hypothesize that capsules can be used effectively for object segmentation with high accuracy and heightened efficiency compared to the state of the art segmentation methods. To show the efficacy of the capsules for object segmentation, we choose a challenging application of pathological lung segmentation from computed tomography (CT) scans.

## 2 Background and Related Works

**Problem definition:** The task of segmenting objects from images can be formulated as a joint object recognition and delineation problem. The goal in object recognition is to locate an object's presence in an image, whereas delineation attempts to draw the object's spatial extent and composition (Bagci et al. [2012]). Solving these tasks jointly (or sequentially) results in partitions of non-overlapping, connected regions, homogeneous with respect to some signal characteristics. Object segmentation is an inherently difficult task; apart from recognizing the object, we also have to label that object at the pixel level, which is an ill-posed problem.

**State-of-the-art methods:** Object segmentation literature is vast, both before and in the deep learning era. Herein, we only summarize the most popular deep learning-based segmentation algorithms. Based on FCN (Long et al. [2015]) for semantic segmentation, Ronneberger et al. [2015] introduced an alternative CNN-based pixel label prediction algorithm, called U-Net, which forms the backbone of many deep learning-based segmentation methods in medical imaging today. Following this, many subsequent works follow this encoder-decoder structure, experimenting with dense connections, skip connections, residual blocks, and other types of architectural additions to improve segmentation accuracies for particular medical imaging applications. For instance, a recent example by Jégou et al. [2017] combines a U-Net-like structure with the very successful DenseNet (Huang et al. [2017]) architecture, creating a densely connected U-Net structure, called *Tiramisu*. As another example, Mortazi et al. [2017b] proposed a multi-view CNN, following this encoder-decoder structure and adding a novel loss function, for segmenting the left atrium and proximal pulmonary veins from MRI. Other successful frameworks for segmentation are SegNet (Badrinarayanan et al. [2017]), RefineNet (Lin et al. [2017]), PSPNet (Zhao et al. [2017]), Large Kernel Matters (Peng et al. [2017]), ClusterNet (LaLonde et al. [2018]), and DeepLab (Chen et al. [2018]).

**Pathological lung segmentation:** Anatomy and pathology segmentation have been central to the most medical imaging applications. Recently, deep learning algorithms have been shown to be

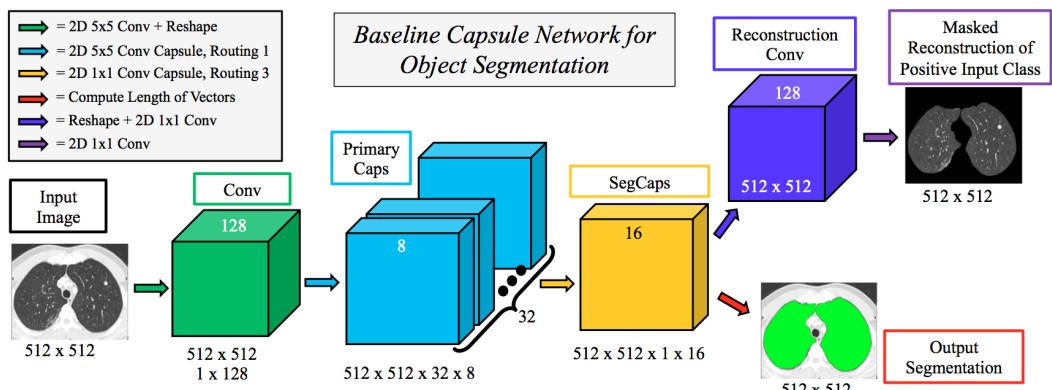

Figure 1: A simple three-layer capsule segmentation network closely mimicking the work by Sabour et al. [2017]. This network uses our proposed locally-constrained dynamic routing algorithm as well as the masked reconstruction of the positive input class.

generally successful for image segmentation problems. Specific to radiology scans, accurately segmenting anatomical structures and/or pathologies is a continuing concern in clinical practice because even small segmentation errors can cause major problems in disease diagnosis, severity estimation, prognosis, and other clinical evaluations. Despite its importance, accurate segmentation of pathological lungs from CT scans remains extremely challenging due to a wide spectrum of lung abnormalities such as consolidations, ground glass opacities, fibrosis, honeycombing, tree-in-buds, and nodules (Mansoor et al. [2014]). In this study, we test the efficacy of the proposed **SegCaps** algorithm for pathological lung segmentation due to precise segmentation's importance as a precursor to the deployment of nearly any computer aided diagnosis (CAD) tool for pulmonary image analysis.

## 3    Building Blocks of Capsules for Image Segmentation

A simple three-layer capsule network showed remarkable initial results in Sabour et al. [2017], producing state-of-the-art classification results on the MNIST dataset and relatively good classification results on the CIFAR10 dataset. Since then, researchers have begun extending the idea of capsule networks to other applications; nonetheless, no work yet exists in literature for a method of capsule-based object segmentation.

Performing object segmentation with a capsule-based network is difficult for a number of reasons. The original capsule network architecture and dynamic routing algorithm is extremely computationally expensive, both in terms of memory and run-time. Additional intermediate representations are needed to store the output of "child" capsules in a given layer while the dynamic routing algorithm determines the coefficients by which these children are routed to the "parent" capsules in the next layer. This dynamic routing takes place between every parent and every possible child. One can think of the additional memory space required as a multiplicative increase of the batch size at a given layer by the number of capsule types at that layer. The number of parameters required quickly swells beyond control as well, even for trivially small inputs such as MNIST and CIFAR10. For example, given a set of 32 capsule types with $6 \times 6$, 8D-capsules per type, being routed to $10 \times 1$, 16D-capsules, the number of parameters for this layer alone is $10 \times (6 \times 6 \times 32) \times 16 \times 8 = 1,474,560$ parameters. This one layer contains, coincidentally, roughly the same number of parameters as our entire proposed deep convolutional-deconvolutional capsule network with locally-constrained dynamic routing which itself operates on $512 \times 512$ pixel inputs.

We solve this memory burden and parameter explosion by extending the idea of convolutional capsules (primary capsules in Sabour et al. [2017] are technically convolutional capsules without any routing) and rewriting the dynamic routing algorithm in two key ways. First, children are only routed to parents within a defined spatially-local kernel. Second, transformation matrices are shared for each member of the grid within a capsule type but are not shared across capsule types. To compensate for the loss of global connectivity with the locally-constrained routing, we extend capsule networks by proposing "deconvolutional" capsules which operates using transposed convolutions, routed by

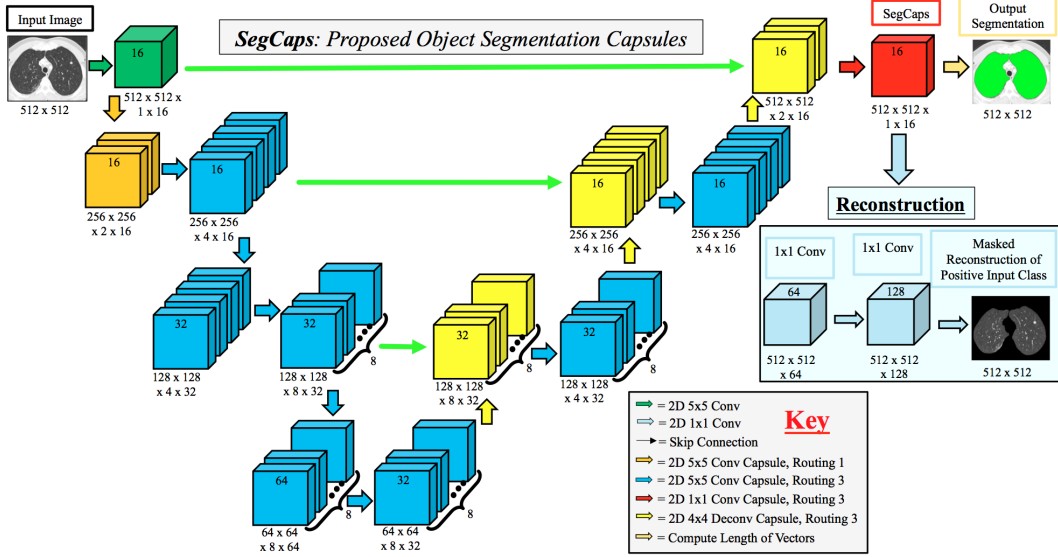

Figure 2: The proposed **SegCaps** architecture for object segmentation.

the proposed locally-constrained routing. These innovations allow us to still learn a diverse set of different capsule types. Also, with the proposed deep convolutional-deconvolutional architecture, we retain near-global contextual information, while dramatically reducing the number of parameters in the network, addressing the memory burden, and producing state-of-the-art results for our given application. Our proposed **SegCaps** architecture is illustrated in Figure 2. As a comparative baseline, we also implement a simple three-layer capsule structure, more closely following that of the original capsule implementation, shown in Figure 1.

## 3.1 Summary of Our Contributions

The novelty of this paper can be summarized as follows:

1. Our proposed **SegCaps** is the first use of a capsule network architecture for object segmentation in literature.

2. We propose two modifications to the original dynamic routing algorithm where (i) children are only routed to parents within a defined spatially-local window and (ii) transformation matrices are shared for each member of the grid within a capsule type.

3. These modifications, combined with convolutional capsules, allow us to operate on large images sizes ($512 \times 512$ pixels) for the first time in literature, where previous capsule architectures do not exceed inputs of $32 \times 32$ pixels in size.

4. We introduce the concept of "deconvolutional" capsules and create a novel deep convolutional-deconvolutional capsule architecture, far deeper than the original three-layer capsule network, implement a three-layer convolutional capsule network baseline using our locally-constrained routing to provide a comparison with our **SegCaps** architecture, investigate two different routing iteration schemes for our **SegCaps**, and extend the masked reconstruction of the target class as a method for regularization to the problem of segmentation as described in Section 4.

5. **SegCaps** produces slightly improved results for lung segmentation on the LUNA16 subset of the LIDC-IDRI database, in terms of dice coefficient, when compared with state-of-the-art methods U-Net (Ronneberger et al. [2015]) and Tiramisu (Jégou et al. [2017]), while dramatically reducing the number of parameters needed to achieve this performance. The proposed **SegCaps** architecture contains $95.4\%$ fewer parameters than U-Net and $38.4\%$ fewer than Tiramisu.

# 4 SegCaps: Capsules for Object Segmentation

As illustrated in Figure 2, the input to our SegCaps network is a $512 \times 512$ pixel image, in this case, a slice of a CT Scan. This image is passed through a 2D convolutional layer which produces 16 feature maps of the same spatial dimensions. This output forms our first set of capsules, where we have a single capsule type with a grid of $512 \times 512$ capsules, each of which is a 16 dimensional vector. This is then followed by our first convolutional capsule layer. We will now generalize this process to any given layer $\ell$ in the network.

At layer $\ell$, there exists a set of capsule types $T^\ell = \{t_1^\ell, t_2^\ell, ..., t_n^\ell \mid n \in \mathbb{N}\}$. For every $t_i^\ell \in T^\ell$, there exists an $h^\ell \times w^\ell$ grid of $z^\ell$-dimensional child capsules, $C = \{\boldsymbol{c}_{11}, ..., \boldsymbol{c}_{1w^\ell}, ..., \boldsymbol{c}_{h^\ell 1}, ..., \boldsymbol{c}_{h^\ell w^\ell}\}$, where $h^\ell \times w^\ell$ is the spatial dimensions of the output of layer $\ell - 1$. At the next layer of the network, $\ell + 1$, there exists a set of capsule types $T^{\ell+1} = \{t_1^{\ell+1}, t_2^{\ell+1}, ..., t_m^{\ell+1} \mid m \in \mathbb{N}\}$. And for every $t_j^{\ell+1} \in T^{\ell+1}$, there exists an $h^{\ell+1} \times w^{\ell+1}$ grid of $z^{\ell+1}$-dimensional parent capsules, $P = \{\boldsymbol{p}_{11}, ..., \boldsymbol{p}_{1w^{\ell+1}}, ..., \boldsymbol{p}_{h^{\ell+1}1}, ..., \boldsymbol{p}_{h^{\ell+1}w^{\ell+1}}\}$, where $h^{\ell+1} \times w^{\ell+1}$ is the spatial dimensions of the output of layer $\ell$.

In convolutional capsules, every parent capsule $\boldsymbol{p}_{xy} \in P$ receives a set of "prediction vectors", $\{\hat{\boldsymbol{u}}_{xy|t_1^\ell}, \hat{\boldsymbol{u}}_{xy|t_2^\ell}, ..., \hat{\boldsymbol{u}}_{xy|t_n^\ell}\}$, one for each capsule type in $T^\ell$. Hence, this set is defined as the matrix multiplication between a learned transformation matrix, $M_{t_i^\ell}$, and the sub-grid of child capsules outputs, $U_{xy|t_i^\ell}$, within a user-defined kernel centered at position $(x, y)$ in layer $\ell$; hence $\hat{\boldsymbol{u}}_{xy|t_i^\ell} = M_{t_i^\ell} \cdot U_{xy|t_i^\ell}, \forall\, t_i^\ell \in T^\ell$. Therefore, we can see each $U_{xy|t_i^\ell}$ has shape $k_h \times k_w \times z^\ell$, where $k_h \times k_w$ are the dimensions of the user-defined kernel. Each $M_{t_i^\ell}$ has shape $k_h \times k_w \times z^\ell \times \mid T^{\ell+1} \mid \times z^{\ell+1}$ for all capsule types $T^\ell$, where $\mid T^{\ell+1} \mid$ is the number of parent capsule types in layer $\ell + 1$. Note that each $M_{t_i^\ell}$ does not depend on the spatial location $(x, y)$, as the same transformation matrix is shared across all spatial locations within a given capsule type (similar to how convolutional kernels scan an input feature map), and this is one way our method can exploit parameter sharing to dramatically cut down on the total number of parameters to be learned. The values of these transformation matrices for each capsule type in a layer are learned via the backpropagation algorithm with a supervised loss function.

To determine the final input to each parent capsule $\boldsymbol{p}_{xy} \in P$, we compute the weighted sum over these "prediction vectors", $\boldsymbol{p}_{xy} = \sum_n r_{t_i^\ell|xy} \hat{\boldsymbol{u}}_{xy|t_i^\ell}$, where $r_{t_i^\ell|xy}$ are the routing coefficients determined by the dynamic routing algorithm. These routing coefficients are computed by a "routing softmax",

$$r_{t_i^\ell|xy} = \frac{\exp(b_{t_i^\ell|xy})}{\sum_k \exp(b_{t_i^\ell k})}, \tag{1}$$

whose initial logits, $b_{t_i^\ell|xy}$ are the log prior probabilities that prediction vector $\hat{\boldsymbol{u}}_{xy|t_i^\ell}$ should be routed to parent capsule $\boldsymbol{p}_{xy}$.

Our method differs from the dynamic routing implemented by Sabour et al. [2017] in two ways. First, we locally constrain the creation of the prediction vectors. Second, we only route the child capsules within the user-defined kernel to the parent, rather than routing every single child capsule to every single parent. The output capsule is then computed using a non-linear squashing function

$$\mathbf{v}_{xy} = \frac{||\boldsymbol{p}_{xy}||^2}{1 + ||\boldsymbol{p}_{xy}||^2} \frac{\boldsymbol{p}_{xy}}{||\boldsymbol{p}_{xy}||}, \tag{2}$$

where $\mathbf{v}_{xy}$ is the vector output of the capsule at spatial location $(x, y)$ and $\boldsymbol{p}_{xy}$ is its final input. Lastly, the agreement is measured as the scalar product $a_{t_i^\ell|xy} = \mathbf{v}_{xy} \cdot \hat{\boldsymbol{u}}_{xy|t_i^\ell}$. A final segmentation mask is created by computing the length of the capsule vectors in the final layer and assigning the positive class to those whose magnitude is above a threshold, and the negative class otherwise. The pseudocode for this locally constrained dynamic routing is summarized in Algorithm 1.

As a method of regularization, we extend the idea of reconstructing the input to promote a better embedding of our input space. This forces the network to not only retain all necessary information about a given input, but also encourages the network to better represent the full distribution of the

---

**Algorithm 1** Locally-Constrained Dynamic Routing.

---

1: **procedure** ROUTING($\hat{\boldsymbol{u}}_{xy|t_i^\ell}, d, \ell, k_h, k_w$)
2:      for all capsule types $t_i^\ell$ within a $k_h \times k_w$ kernel centered at position $(x, y)$ in layer $\ell$ and capsule $xy$ centered at position $(x, y)$ in layer $(\ell + 1)$: $b_{t_i^\ell|xy} \leftarrow 0$.
3:      **for** $d$ iterations **do**
4:          for all capsule types $t_i^\ell$ in layer $\ell$: $\mathbf{c}_{t_i^\ell} \leftarrow \texttt{softmax}(\mathbf{b}_{t_i^\ell})$      $\triangleright$ `softmax` computes Eq. 1
5:          for all capsules $xy$ in layer $(\ell + 1)$: $\boldsymbol{p}_{xy} \leftarrow \sum_n r_{t_i^\ell|xy} \hat{\boldsymbol{u}}_{xy|t_i^\ell}$
6:          for all capsules $xy$ in layer $(\ell + 1)$: $\boldsymbol{v}_{xy} \leftarrow \texttt{squash}(\boldsymbol{p}_{xy})$      $\triangleright$ `squash` computes Eq. 2
7:          for all capsule types $t_i^\ell$ in layer $\ell$ and capsules $xy$ in layer $(\ell + 1)$: $b_{t_i^\ell|xy} \leftarrow b_{t_i^\ell|xy} +$
$\hat{\boldsymbol{u}}_{xy|t_i^\ell} . \mathbf{v}_{xy}$
         **return** $\mathbf{v}_{xy}$

---

input space, rather than focusing only on its most prominent modes. Since we only wish to model the distribution of the positive input class and treat all other pixels as background, we mask out segmentation capsules which do not belong to the positive class and reconstruct a similarly masked version of the input image. We perform this reconstruction via a three layer $1 \times 1$ convolutional network, then compute a weighted mean-squared error loss between only the positive input pixels and this reconstruction.

## 5 Experiments and Results

Experiments were conducted on the LUNA16 subset of the LIDC-IDRI database, randomly split into four training/testing folds for performing k-fold cross-validation. The LUNA16 subset contains a range of lung CT scans from severe to no pathologies present. Ground-truth annotations were provided in the form of segmentation masks created by an automated algorithm (van Rikxoort et al. [2009]). Manual inspection led to the removal of 10 of the 888 CT scans due to exceedingly poor annotations. Because of the lack of expert human-annotations, we observed that the proposed methods and baselines usually outperformed these ground-truth segmentation masks for particularly difficult scans. This, in turn, lead to higher dice scores for worse performance in those cases, as they typically failed in a similar way. To compensate for such outliers, all numeric results are reported in terms of median rather than mean averages.

U-Net, Tiramisu, our three-layer baseline capsule segmentation network, and **SegCaps** are all implemented using Keras with TensorFlow. For the baseline capsule network, we modify the margin loss from Sabour et al. [2017] to the weighted binary version. All other methods are trained using the weighted BCE loss for the segmentation output. We note that in small-scale experiments, the weighted margin loss seemed to perform comparable to the weighted BCE loss for **SegCaps**. Yet, more thorough experiments are needed to draw conclusions from this. The reconstruction output loss is computed via the masked MSE as described in Section. 4. All possible experimental factors are controlled between different networks. All networks are trained from scratch, using the same data augmentation methods (scale, flip, shift, rotate, elastic deformations, and random noise) and Adam optimization (Kingma and Ba [2014]) with an initial learning rate of $0.00001$. A batch size of 1 is chosen for all experiments to match the original U-Net implementation. The learning rate is decayed by a factor of $0.05$ upon validation loss stagnation for $50,000$ iterations and early stopping is performed with a patience of $250,000$ iterations based on validation dice scores.

The final quantitative results of these experiments are shown in Table 1. **SegCaps** slightly outperforms all other compared approaches with an average dice score of $98.479\%$, while requiring far fewer parameters, a reduction in parameters of **over 95**$\%$ from U-Net and **over 38**$\%$ compared with Tiramisu. For qualitative evaluations, we have shown two different slices from two different CT scan and highlighted the segmentation leakages that U-NET caused in Fig. 3.

Further, we investigate how different capsule vectors in the final segmentation capsule layer are representing different visual attributes. Figure 4 shows the selected 5 visual attributes (each row) out of 16 (dimension of final capsule segmentation vector) across different values of the vectors (each column). We observe that regions with different textural properties (i.e., small and large homogeneous) are progressively captured by the capsule segmentation vectors.

Table 1: Dice Coefficient results on a 4-fold cross-validation split of the LUNA16 dataset. For **SegCaps** (R1), dynamic routing is only performed on layers which change spatial dimensions. All other layers are routed with equal-weight coupling coefficients.

| Method | Parameters | Split-0 (%) | Split-1 (%) | Split-2 (%) | Split-3 (%) | Average (%) |
|---|---|---|---|---|---|---|
| U-Net | 31.0 M | 98.353 | 98.432 | 98.476 | **98.510** | 98.449 |
| Tiramisu | 2.3 M | 98.394 | 98.358 | **98.543** | 98.339 | 98.410 |
| Baseline Caps | 1.7 M | 82.287 | 79.939 | 95.121 | 83.608 | 83.424 |
| SegCaps (R1) | **1.4 M** | 98.471 | 98.444 | 98.401 | 98.362 | 98.419 |
| SegCaps | **1.4 M** | **98.499** | **98.523** | 98.455 | 98.474 | **98.479** |

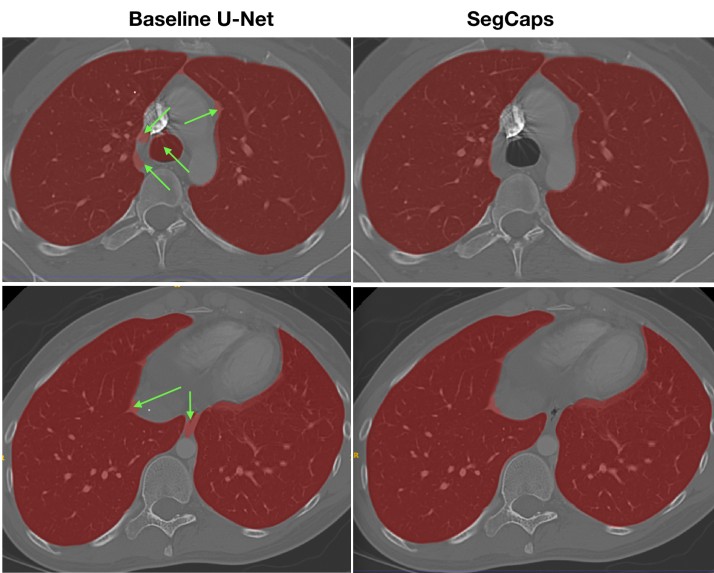

Figure 3: Qualitative results for U-Net and the proposed **SegCaps** on two different slices from CT scans. The left column is the results produced by U-Net. The right column is the results produced by the proposed **SegCaps** algorithm on these same slices. The green arrows highlight areas where U-Net made errors in segmentation.

## 6 Conclusion

We propose a novel deep learning algorithm, called **SegCaps**, for object segmentation, and showed its efficacy in a challenging problem of pathological lung segmentation from CT scans. The proposed framework is the first use of the recently introduced capsule network architecture and expands it in several significant ways. First, we modify the original dynamic routing algorithm to act locally when routing children capsules to parent capsules and to share transformation matrices across capsules within the same capsule type. These changes dramatically reduce the memory and parameter burden of the original capsule implementation and allows for operating on large image sizes, whereas previous capsule networks were restricted to very small inputs. To compensate for the loss of global information, we introduce the concept of a deep convolutional-deconvolutional capsule architecture for pixel level predictions of object labels. Finally, we extend the masked reconstruction of the target class as a regularization strategy for the segmentation problem. Experimentally, **SegCaps** produces slightly improved accuracies for lung segmentation on the LUNA16 subset of the LIDC-IDRI database, in terms of dice coefficient, when compared with state-of-the-art networks U-Net (Ronneberger et al. [2015]) and Tiramisu (Jégou et al. [2017]). More importantly, the proposed **SegCaps** architecture contains $95.4\%$ fewer parameters than U-Net and $38.4\%$ fewer than Tiramisu. The proposed algorithm fundamentally improves the current state-of-the-art object segmentation approaches, and provides strong evidence that capsules can successfully model the spatial relationships of the objects better than traditional CNNs.

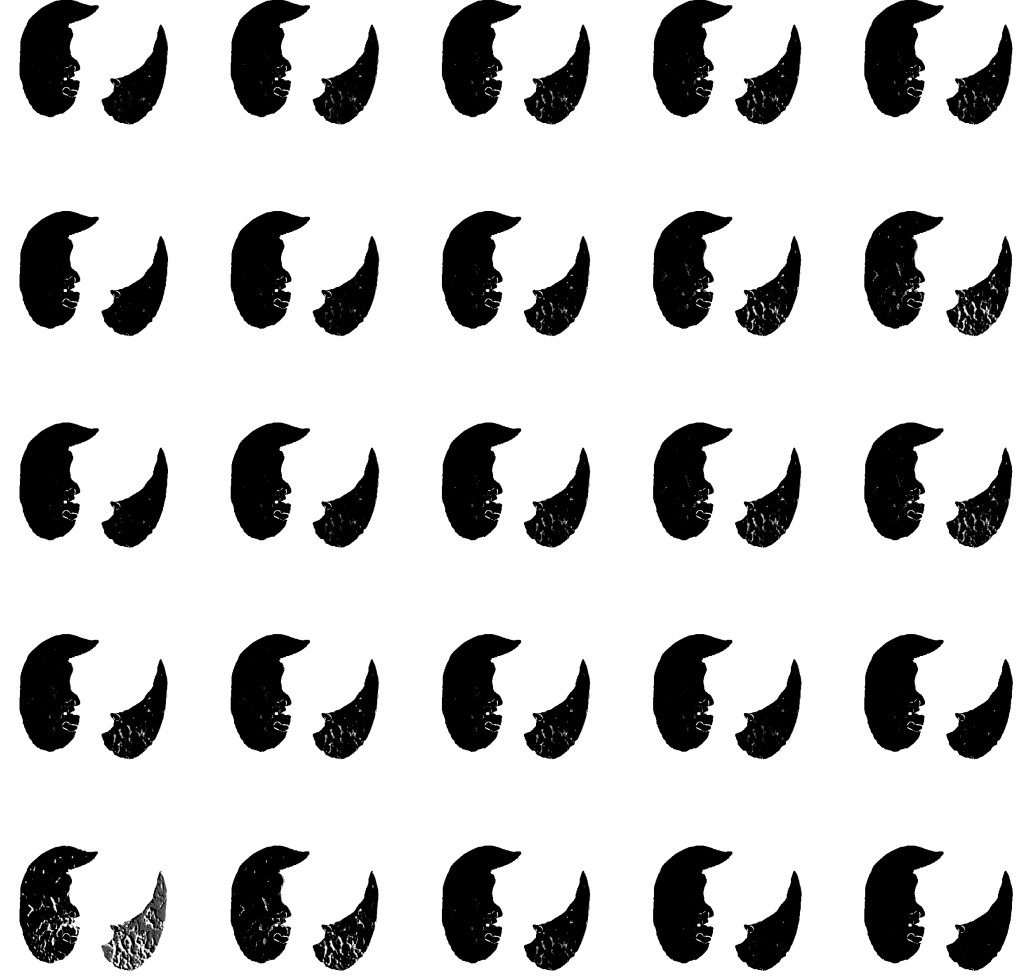

Figure 4: Each row includes 5 selected visual attributes from 16 dimensional capsule segmentation vectors. Each column, on the other hand, indicate different visual attributes of these vectors for a given interval obtained from the capsule segmentation vector.

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
