# OpenReview forum: "Capsules for Object Segmentation"
_MIDL.amsterdam/2018/Conference — MIDL 2018 Oral_

### Review · AnonReviewer3 · 2018-05-06
**Nice extension of capsules for segmentation but not very convincing evaluation**

**Rating:** 4
**Confidence:** 3

**Review:**

Authors present an extension of the capsule networks, which was originally published in 2017, for object segmentation. To the best of the knowledge of this reviewer, this is the first article that applies capsule networks to image segmentation.
Pros:
1. This is the first application of capsules to image segmentation. In that respect, the contribution is original and introduces an interesting avenue for image segmentation research.
2. Authors propose a simple yet effective way of extending capsules for the segmentation problem without increasing the memory requirement and the number of parameters by a large margin.
3. The explanation of the method somewhat relies on the original capsule network paper. However, despite that the method is clearly explained and the text is well written.

Cons:
1. The motivation for using capsules for object segmentation is not explicitly made. Authors mention the advantages of capsules for object detection but these are rewritten from the original work.
2. The evaluation is the weakest spot of this article in my opinion. The following points reduces enthusiasm for the article:
2a. Authors claim that the proposed structure reduces network parameters compared to U-Net and Tiramisu. This statement is rather vague. Authors did not perform any experiments with U-Net and Tiramisu using similar parameters as SegCaps. Segmentation performance of these networks with low number of parameters would be useful to back-up the claim of reducing required parameters.
2b. There are two components of the proposed network that are hard to disentangle. First component is the capsules and the convolutional-deconvolutional extensions. Second component is the regularisation by reconstruction. The same regularisation used with the other networks might also yield similar accuracy improvements.
2c. The improvements in Dice scores compared to U-Net and Tiramisu are really small and shows variation in the cross-validation experiments. I understand that large structures, small Dice improvements can indicate large local differences. However, 0.0003 improvement is rather small. This also questions the examples shown in Figure 3. How do the errors of the SegCaps look like? Given these quite modest — assuming they are not just due to chance — improvements, I think it is difficult to conclude that the proposed algorithm “fundamentally improves” the current state-of-the-art.

Overall, I think it is nice idea to extend capsules to image segmentation. Article is well written and the method is clearly explained. Evaluation is the main weakness and does not support the conclusions authors would like to reach in my opinion. Despite the problems, it would be very nice to see this article in the conference and to discuss with the authors.

**Special Issue:**

No

---

### Review · AnonReviewer1 · 2018-05-07
**Some novelty, some things to be desired**

**Rating:** 4
**Confidence:** 3

**Review:**

Overall, the contribution as presented is more relevant to the advancement of neural network architectures and methods than it is to the field of medical image segmentation.

* Pro: The approach is thoughtful, novel, and will probably lead to subsequent papers that will qualify the proposed methods on more data and more difficult problems (like multi-class/multi-label segmentation, 3D volumetric approaches).
* Pro: The paper is well-written and exhaustive in the description of its implementation, and as soon as the code is released (which has been promised), it will help other researchers transfer the approach to own data.

* Con: In particular for the U-Net, there are various ways to reduce parameters, the simplest being to use fewer layers. What would performance have looked like then? Isn't the interesting question what is gained by the novel architecture against comparably large (in terms of parameters) other architectures?
* Con: The mere average dice score over hundreds of cases doesn't tell you much if it only improves by so little. It would be more interesting to see the distribution over cases or some other presentation to better understand the systematics of improvement. From the numbers in Table 1 and comparison in Fig. 3 I would suspect that in other cases, there are counter examples where the U-Net is "better".

* Remark: The arrows in Fig. 2 "shortcut connections" are differently coloured in legend and graphics.

**Special Issue:**

Yes

---

### Review · AnonReviewer2 · 2018-05-09
**The authors extend capsule networks to the case of semantic segmentation, in an efficient way, by limiting the neighborhood of capsules that can be children of another one (also sharing transformation matrices).**

**Rating:** 3
**Confidence:** 2

**Review:**

The main contribution of this paper is to propose an alternative semantic segmentation net, based on capsules and so, different from standard networks, while leading to competitive results.

The authors should rephrase some statements such as "The proposed algorithm fundamentally improves the current state-of-the-art object segmentation approaches, and provides strong evidence that capsules can successfully model the spatial relationships of the objects better than traditional CNNs". There is no evidence of this in the experimental section, which in itself presents some weaknesses:
First, because only one dataset is used.
Second, because the differences reported in the performance look minimal between the proposed model and the competitors, and it is unclear if they are significant.
Third, it is not clearly motivated why reconstruction is needed as a means for regularization. It is also unclear whether the competitors use the same regularization method or not. In any case, it would be desirable to have an experiment to check the effect of such regularization method.

The paper reads ok, although some sentences are difficult to parse (also there is a weird use of the exist symbol in Sec. 4).
In general, it is an interesting paper with promising results, but the empirical evidences are far from what it is claimed in the conclusions.

**Special Issue:**

Yes

---

### Comment · ~Dejan_Štepec1 · 2018-04-18
**Review and some questions, especially about section 4**

Hi,

first of all thank you for the great work that you all have done. I think this is one of the first papers dealing with Capsules that goes beyond the initial architecture and ideas presented in NIPS paper. It's also one of the first papers dealing with Capsules that is solving some real life problem on a dataset beyond simple MNIST etc. Though I do have some complaints about the dataset used.

Sections 1-3 seems to be fine. You have described the idea and the background in a sufficient manner. Related work is maybe written a bit to short as it doesn't represent the past work in more details. The main drawback of this paper is that it focuses to much on this particular problem of lung segmentation. You presented the idea that should be tested more thoroughly on other domains also in order to present the generalization possibility. I don't mean that you should test the method on COCO segmentation dataset (which would be nice, would need to be extended for color images and multiple classes, which shouldn't be hard and I also don't expect to be particularly good there but still maybe it would show that there is potential). At least you should find some other datasets that could prove that the method does work also for other problems and no only for the lung segmentation on this particular dataset where it seems that the only error that is left is probably because of the errors in annotations.

Section 4 describes the details of the implementation and I was having quite a few problems understanding in details how the things are implemented.

- Each parent capsule "p_xy" receives a set of predictions vectors "u_xy_ti", one for each capsule type t_i: As I understand,  for the parent capsule "p_xy" we are considering all the child capsules layer before around the position "xy" within the kernel size (for the case where spatial dimensions of the layer l+1 are the same as layer l) for each capsule type separately. As you described it with the notation (set notation), it seems like every parent capsule gets only 1 prediction vector from each capsule type, instead we get a set of prediction vectors (from capsules within kernel size) from every capsule type for each location p_xy.

- Transformation matrix M: Just for the clarification, transformation is not operated as convolutions? (from the definition of the dimensions of "M" I could answer myself on the question above, that there is not only 1 predicted vector for each location). Instead we do regular matrix multiplication here and for each operation M x U we get (k_h * k_w * |capsule types in l +1|)  z_(l+1) dimensional vectors for each capsule type which then used in Routing...?

- When there is a change of spatial dimensions between layers you probably compute the matrix multiplications to get prediction vectors with stride of 2 or you use some other strategy of interpolation between capsule vectors?

- Locally-Constrained Dynamic Routing: Again there is some misunderstanding caused by the notation. I suspect that with the notation b_ti|xy you mean all the capsules in capsule type t_i within the kernel size. So that every child capsule in a window in each capsule type has it's own weight that is learnt. Notation is ambiguous and one may think that that there is only 1 coefficient learnt for every "capsule type". The difference and proper use and especially distinction should be made between "capsule types" and "capsules" as often in the paper everything is generalized to the capsule types like in this case where you think that all capsules in a window in a capsule type have the same weight.

- As one the main contributions you mention "deconvolutional capsules", you never describe it in more detail. I think that it deserves subsection in section 4 for the sake of completeness.

- I am missing loss description here also. You mention it in section 5 -> "weighted binary version". This deserves at least equation form to be presented.

I have no complaints about the section 5. Just the experiments should be done on different datasets.

- Are you planning to publish the code? If yes, when?

Anyways I like the idea as it represents quite simple extension of the initial CapsNets in a much more scalable way that can be used on real-life domains. Given the targeted conference and quite a few novelties I think that the paper should be accepted with some modifications to the language and descriptions that would clarify some of the ambiguities in the paper.

---

> ### Comment · ~Rodney_LaLonde1 · 2018-04-21
> **Response to Dejan Štepec**
>
> Hello Dejan Štepec,
>
> Thank you for taking the time to read and review our work, and for the kind words! We appreciate your constructive comments as well and will address them in the order your mentioned them.
>
> - “The main drawback of this paper is that it focuses to much on this particular problem of lung segmentation.”
> Since MIDL is a conference with medical image emphasis, we focused on the task of lung segmentation, which is a challenging and important problem in medical image computing. Showing results on MS COCO or other computer vision datasets would be inappropriate for this conference. On the other hand, we agree with you about expanding the impact of our work with other vision data set. We are happy to inform you that we plan to present our computer vision results soon at a different venue, extending our method to color images and multi-class for performing semantic segmentation in a natural image setting.
>
> - “Each parent capsule "p_xy" receives a set of predictions vectors "u_xy_ti", one for each capsule type t_i... it seems like every parent capsule gets only 1 prediction vector from each capsule type, instead we get a set of prediction vectors (from capsules within kernel size) from every capsule type for each location p_xy...”
> - “Transformation matrix M: Just for the clarification, transformation is not operated as convolutions?”
> - “Locally-Constrained Dynamic Routing: Again there is some misunderstanding caused by the notation.”
> We apologize for the minor confusion in this section. For convolutional capsules, the transformation is computed as a dot product between M and U. Each prediction vector u_xy for a given capsule type is the result of the dot product between M and U_xy, resulting in a single value to be routed for each capsule type while still taking into account local context information from surrounding capsules. Note that we can keep this a cross product rather than a dot product and route each resulting prediction vector, but this would result in k_h x k_w times the needed memory at each layer to store these values. We will be sure to change M x U to M *(dot) U as this was a minor error in notation which led to the confusion. The rest of the language of a single prediction vector is correct and should clear up all confusion.
>
> - "When there is a change of spatial dimensions... stride of 2...?"
> Yes this is correct, layers which change spatial dimension have a stride of two in our architecture. This is true as well for the deconvolutional capsules where the transposed convolutions have stride (2,2).
>
> - "... 'deconvolutional capsules', you never describe it in more detail... deserves subsection in section 4 for the sake of completeness."
> Deconvolutional capsules operate on transposed convolutions to form their prediction vectors, then these vectors are routed via the locally-constrained dynamic routing algorithm. A few sentences about this could be included for completeness. We plan to devote a small subsection in section 4 to explain these further.
>
> - “I am missing loss description here...”
> We did not feel the loss function was sufficiently different from the one presented in Sabour et al. so we did not include it; however, we can add this equation where the loss is mentioned.
> L = Sum_x[Sum_y[[alpha * T_xy * max(0, m^+ - ||v_xy||)^2 + lambda * (1 - T_xy) * max(0, ||v_xy|| - m^-)^2]]
> where alpha is a class balancing term of the ratio of negative to positive class pixels in the training data, x and y are spatial locations, T_xy is 1 iff the pixel at (x,y) belongs to the positive class, m^+ = 0.9, m^- = 0.1, and lambda = 0.5.
> For the weighted binary cross-entropy formulations, we use the standard formulation in tensorflow using the same class balancing term described above. We found that using the weighted cross entropy or the weighted margin loss did not produce significantly different results from each other.
>
> - "Are you planning to publish the code? If yes, when?"
> Yes! We are cleaning up the code right now and plan to have it made public within the next week.

---

### Comment · ~Rodney_LaLonde1 · 2018-05-09
**Public Code Release**

I am happy to announce we have made the code for this project public. It can be found at https://github.com/lalonderodney/SegCaps.

---

### Decision · Program_Chairs · 2018-05-15
**Paper95 Acceptance Decision**

Oral